# Rosen-Type Piezoelectric Transformers Based on 0.5Ba(Zr_0.2_Ti_0.8_)O_3_–0.5(Ba_0.7_Ca_0.3_)TiO_3_ Ceramic and Doped with Sb_2_O_3_

**DOI:** 10.3390/ma16186201

**Published:** 2023-09-14

**Authors:** Suk-Young Jung, Gwangseop Lee, Tae-wan Kim, Sung-Jin Kim, Jung-Hyuk Koh

**Affiliations:** 1School of Electrical and Electronic Engineering, Chung-Ang University, Seoul 06974, Republic of Korea; mmf416@naver.com; 2Department of Intelligent Energy and Industry, Chung-Ang University, Seoul 06974, Republic of Korea; 3College of Electrical and Computer Engineering, Chungbuk National University, Cheongju 28644, Republic of Korea

**Keywords:** piezoelectric transformers, 0.5Ba(Zr_0.2_Ti_0.8_)O_3_−0.5(Ba_0.7_Ca_0.3_)TiO_3_, soft-doped Sb_2_O_3_, Rosen-type, step-up amplification ratio

## Abstract

In this study, the characteristics in the lead-free piezoelectric ceramic 0.5Ba(Zr_0.2_Ti_0.8_)O_3_−0.5(Ba_0.7_Ca_0.3_)TiO_3_ (0.5BZT–0.5BCT) were investigated to assess its potential for Rosen-type piezoelectric transformers. This piezoelectric ceramic has a piezoelectric charge coefficient d_33_ of 430 pC/N, an electromechanical coupling factor k_p_ of 49%, a dielectric constant ε_r_ of 2836, a remnant polarization P_r_ of 4.98 μC/cm^2^, and a coercive electric field E_c_ of 2.41 kV/cm. Sb_2_O_3_ was soft doped with 0.05, 0.1, 0.15, and 0.2 mol%, respectively, and exhibited excellent physical properties at 0.1 mol%. Based on this, a piezoelectric transformer was fabricated and measured, and it showed better output characteristics than pure 0.5BZT–0.5BCT. The amplification ratio (V_out_/V_in_) was optimized based on the device geometry and properties of the piezoelectric material. Moreover, the output characteristics of the Rosen−type piezoelectric transformer were simulated with the PSpice program. Output values of the fabricated and simulated piezoelectric transformers for the r vibrational frequency were compared and analyzed. Accordingly, the step-up amplification ratios V_out_/V_in_ of the fabricated and simulated devices at the vibrational frequency were compared as well. This piezoelectric transformer could replace silicon steel transformers and be used for the creation of black light and for laptop chargers.

## 1. Introduction

The Curie brothers discovered piezoelectricity in 1880 [1]. Researchers have mainly studied lead-based materials such as the piezoelectric ceramic Pb(Zr,Ti)O_3_ (PZT) [2]. Owing to their high electro-mechanical coupling coefficients, they have been used in different applications including ultrasonic motors and transformers. From 1960 to 1962, the General Electric team in Syracuse continued Rosen and Tehon’s work on ceramic power transformers with a research agreement granted by the United States Navy [3]. A member of General Electric Co.’s Electronics Laboratory, edited the book “Solid State Magnetic and Dielectric Devices” authored by Rosen (then at the Stanford Research Institute in Menlo Park, CA, USA) and Tehon. The publication was pioneering work that comprehensively delved into the intricacies of piezoelectric transformer technology, remaining a significant point of reference for researchers in the field [4,5]. As more and more portable electronic devices become smaller and lighter, piezoelectric transformers (PZTs) are becoming widely available. A PZT is essentially an electrical energy transmission device, which appeared in the 1950s of the 20th century. A PZT exchanges energy through electromechanical coupling between adjacent piezoelectric actuators and transducers. However, PZTs have been considered to be environmentally hazardous because they contains lead. Because of that, we researched environmentally harmless lead-free materials [6,7,8]. Current lead-free piezoelectric systems include sodium potassium niobate (K,Na)NbO3 (KNN), sodium bismuth titanate (Na0.5Bi0.5)TiO3 (NBT), bismuth ferrite BiFeO3 (BF), and barium titanate BaTiO3 (BT) ferroelectric systems [9]. In this study, we employed the piezoelectric material 0.5Ba(Zr_0.2_Ti_0.8_)O_3_−0.5(Ba_0.7_Ca_0.3_)TiO_3_ (0.5BZT−0.5BCT) due to the fact that 0.5BZT–0.5BCT demonstrates comparable piezoelectric properties to systems related to PZTs [10]. Acceptor doping produces hard materials, whereas donor doping yields soft materials. Hard piezoelectric substances can withstand significant electrical and mechanical pressures, while soft piezoelectric materials are typically chosen for applications requiring high coupling coefficients and heightened charge sensitivity. Sb_2_O_3_, a soft material, has an increasing effect on the piezoelectric constant(d_33_), coupling coefficient(k_p_), permittivity(ε_r_), and P−E loop of BZT−BCT [11,12].

Over the past few decades, the utilization of piezoelectric resonators for converting AC signal parameters has emerged as an intriguing and promising avenue for exploration. A piezoelectric transformer is essentially a straightforward vibrating apparatus consisting of a resonator with separate electrodes for practical application. When an AC voltage is applied to the input electrode, it induces mechanical deformation across the entire piezoelectric resonator through the converse piezoelectric effect. The charge generated by the piezoelectric transformer’s piezoelectric transfer effect is directly utilized by the collecting output electrodes. These AC voltages are amplified as well by driving a piezoelectric transformer using mechanical vibration [13,14,15]. Piezoelectric transformers have been extensively studied because of their high-power densities, small sizes, low weights, and high efficiency. Since Rosen’s invention of the piezoelectric transformer in 1957, the application field has gradually been expanded and researchers still study its use under practical conditions [4,16,17]. A typical transformer based on magnetic materials comprises a coil wound around a silicon steel sheet core; the voltage is converted according to the ratio of the numbers of turns of the primary and secondary coils. Magnetic transformers convert electrical energy into magnetic energy and subsequently transform this magnetic energy into electrical output. Magnetic transformers are huge because of the magnetic coils and passivation silicon oil. Piezoelectric transformers can also convert energy into a voltage output. However, they are small, do not contain oil containers, and respond fast. Piezoelectric transformers require high-power amplification with low-output impedances and high-energy efficiency. For this amplification, the piezoelectric properties must be improved. Moreover, the voltage of the natural vibration frequency on the secondary side vibrates in a certain direction; this vibration provides the converted secondary voltage at the output. When the secondary voltage is provided, it is affected by the piezoelectric characteristics, The electromechanical coupling factors(k_31_, and k_33_), mechanical quality factor (Q_m_), elastic compliance constants ($s_{33}^{E}$, $s_{11}^{E}$, and $s_{33}^{D}$), piezoelectric transformer output length (L_out_), and thickness(t) can be arranged to make the output formula for this piezoelectric transformer as follows in Equation (1) [13].
(1)$$|\frac{V_{out}}{V_{in}}|=\frac{8}{\pi^{2}}k_{31}k_{33}Q_{m}\frac{\sqrt{s_{33}^{E}/s_{11}^{E}}}{1+\sqrt{s_{33}^{D}/s_{11}^{E}}}\frac{L_{out}}{t}$$

If the output portion exceeds the input portion, the transformer is the step-up transformer; if it is lower than the input portion, the transformer is the step-down transformer [18,19]. This Rosen-type piezoelectric transformer does not require a magnetic field and conduction current during operation. Therefore, piezoelectric transformers do not exhibit a leakage magnetic flux or heating loss during operation. They are highly efficient and have simple structures, small sizes, and light weights [20,21]. Piezoelectric transformers can be supplied for cell phone battery chargers, fluorescent lamp lighting, and LCD backlight driving circuits. Their application field is increasing [4,13,22,23].

In this study, the characteristics of 0.5BZT–0.5BCT were studied to assess its potential for lead-free piezoelectric transformers. Its excellent piezoelectric features (ε_r_ = 2860, k_p_ = 42.4%, d_33_ = 430 pC/N, E_c_ = 6.5 kV/cm, and P_r_ = 0.5 µC/cm^2^) make it suitable for transformers. In addition, the characteristics of Sb_2_O_3_ doped with 0.1 mol% displays excellent piezoelectric characteristics (ε_r_ = 4040, k_p_ = 55.3%, d_33_ = 581 pC/N, E_c_ = 4.5 kV/cm, and P_r_ = 1.2 µC/cm^2^) and its softness makes it suitable for transformers [12]. The output voltage was predicted with circuit simulations, and the electrical output was measured and compared with the simulated value.

## 2. Materials and Methods

Piezoelectric discs were manufactured using a conventional solid-state reaction method, employing ceramic powders composed of 0.5Ba(Zr_0.2_Ti_0.8_)O_3_-0.5(Ba_0.7_Ca_0.3_)TiO_3_. The starting materials were BaCO_3_ (purity 99.9%, Sigma-Aldrich Co., Ltd., St. Louis, MO, USA), CaCO_3_ (Immaculateness 98.9%, Sigma-Aldrich Co., Ltd.), TiO_2_ (Immaculateness 99.6%), ZrO_2_ (Immaculateness 98.9%), and Sb_2_O_3_ (Immaculateness 99.7%) powders. These ceramic powders were weighed according to the stoichiometric ratio. Subsequently, ethyl alcohol and zirconia balls were added to the weighed powder, which was then ball-milled for 24 h using a stabilize zirconia ball. After drying at 120 °C for 12 h, required amounts of the Sb_2_O_3_ (0.05, 0.1, 0.15, 0.2 mol%) and the 0.5Ba(Zr_0.2_Ti_0.8_)O_3_−0.5(Ba_0.7_Ca_0.3_)TiO_3_ ceramic powders were calcined at 1350 °C for 2 h. In addition, polyvinyl alcohol (a binder) was added, and a force of four metric tons was supplied to form a disk. The produced compacts were sintered via heating at 3 °C/min to 1475 °C for 3 h. Moreover, Ag paste was applied to the lower and upper surfaces of the disk-shaped sample, which was then heated to 700 °C for 10 min [17]. At normal room temperature, these samples were immersed in silicone oil and exposed to an electric field of 3.5 kV/mm for 30 min during the poling procedure. The dielectric constant (ε_r_) was measured with an impedance analyzer (Agilent 4294 A, Leasametric 4 Avenue de Norvège A) at 1 kHz, and Ferroelectric hysteresis loops were obtained at a frequency of 100 Hz. These measurements were conducted on disk-shaped samples that had been previously polarized using an electric field of 3.5 kV/cm, and a TReK Technologies high-voltage amplifier (Trek 609E-6-4000V) was utilized for this purpose [22]. Furthermore, the impedance analyzer was used to determine the electromechanical coupling factor (k_p_) under normal temperature conditions [4].

After going through the same process from basis weight to calcination as in the disk manufacturing process, to fabricate piezoelectric transformers, the piezoelectric ceramic was inserted into a specially designed mold. After sintering, the specimens were polished to samples with sizes of 9 × 4.5 × 1.5 mm^3^; Ag paste was applied on the upper half, lower half, and both sides. In the next step, poling of ceramics was accomplished through a two-step process. The samples were heated to 700 °C for 10 min to create electrodes. These samples were subjected to polarization in silicon oil at an ambient temperature, using a DC electric field of 1.5 kV/mm along the longitudinal axis (output section) and 3.5 kV/mm through the thickness (input section) at a normal temperature for 30 min. After 24 h, the electrical characteristics of the piezoelectric transformers were measured with an oscilloscope (DSO-X 2002A), which is a function generator (Keysight 33500B) [22].

## 3. Results and Discussion

Figure 1 presents a schematic diagram of a Rosen−type piezoelectric transformer and its sample size and poling direction. The left half (4.5 mm length) of the transformer was poled in the vertical direction, and the other portion was poled in the horizontal direction. The prepared piezoelectric transformer is a step-up transformer: it vibrates in the longitudinal mode with vertical polarization in the input portion and longitudinal polarization in the output portion [4]. This Rosen-type piezoelectric transformer is highly efficient when the input electrical voltage has a frequency near its vibrational frequency. The function of the piezoelectric transformer generates strong vibration in the longitudinal direction by the reverse piezoelectric effect when the input voltage of the natural vibrational frequency, determined by the total length 2 L, which is expressed as the sum of the length L of the driving portion and the length L of the power generating portion, is supplied to the driving portion.

Figure 2 presents a schematic diagram of a features arrangement for the piezoelectric transformer. The function generator generates an input signal, which is supplied to the V_in_ electrode of the piezoelectric transformer through the amplifier to increase the input signal. The input voltage induces vibrations in the piezoelectric transformer [13]. When an AC voltage is supplied to the ceramic’s halfway point, it initiates continuous longitudinal oscillations at the same frequency as the provided voltage, which demonstrates the inverse piezoelectric effect. These vibrations are mechanically linked to the terminal areas, resulting in the generation of a potential difference, exemplifying the direct piezoelectric phenomenon. In essence, the input zone serves as an actuator, while the output area acts as a transducer. If the oscillation frequency is precisely a multiple of the half-wavelength of the piezoelectric transformer, the transformer begins mechanical vibration [24]. The amplified voltage output from the piezoelectric transformer can be quantified with the use of an oscilloscope. In this way, we can ascertain the step-up amplification ratio; it is related to the geometry of the piezoelectric transformer and its piezoelectric properties [13].

As seen in Figure 3a, the capacitance and dielectric constant (ε_r_) were measured with an impedance analyzer (Agilent 4294 A) at 1 kHz according to Sb_2_O_3_ dopant (0.0, 0.05, 0.1, 0.15, 0.2 mol%) 0.5BZT−0.5BCT and, as seen in Figure 3b, the P-E loop supplied the voltage at 3.5 kV/mm at the normal temperature of the 0.5BZT–0.5BCT material and material doped with Sb_2_O_3_ 0.1 mol% was measured. Through this experiment, the use of the capacitance and dielectric constant was able to find the optimal dopant composition. As a result, the best characteristics were shown at Sb_2_O_3_ 0.1 mol%, and, in (b), The P-E loop voltage was supplied at 3.5 kV/mm and measured at the normal temperature. The P-E loop was measured to confirm the characteristics of softness with high P_r_ and small E_c_. As a result of measuring the piezoelectric properties, the following Table 1 was made. This is a soft characteristic and it can exhibit better results in the output characteristics of a piezoelectric transformer [25,26,27]. Figure 3c shows the 0.5BZT−0.5BCT ceramic by using a scanning electron microscopy image. It can be seen that it has a uniform particle size but also a narrow density microstructure. As seen in Figure 3d, the particle size and density microstructure of ceramic doped with 0.1 mol% Sb_2_O_3_ in 0.5BZT−0.5BCT were expanded. These properties of piezoelectric ceramics tend to increase the dielectric constant and capacitance. Therefore, it is predicted that 0.1 mol% Sb_2_O_3_ dopant will increase the properties in the piezoelectric transformer.

Figure 4 shows that all the samples consisted solely of the perovskite phase, with no presence of any secondary phase. The X-ray diffraction analysis revealed characteristic peaks corresponding to the perovskite crystal structure at orientations (100), (110), (002)/(200), and (210). The presence of a solely perovskite phase in the ceramic systems signaled the effective incorporation of Sb ions into the 0.5BZT−0.5BCT lattice. Furthermore, the inclusion of 0.5BZT−0.5BCT was found to enhance the stability of the perovskite phase and to improve the uniformity of the solid solution. In particular, the doping of Sb at 0.1 mol% revealed a splitting of the peak corresponding to the (002)/(200) reflection. This outcome indicates the coexistence of both the tetragonal and rhombohedral phases in the ceramic material [12].

Figure 5 shows the matching circuit of 0.5BZT−0.5BCT ceramics, with and without doping with a Sb_2_O_3_ 0.1 mol% piezoelectric transformer. The matching circuit was modeled by considering the input portion and output portion of the piezoelectric transformer separately [21,28]. In addition, the internal passive components such as the capacitor, resistor, and inductor were considered. Test signals were supplied to the piezoelectric transformers to extract proper parameters in the RLC passive components. The parameters C_1_, R_1_, and L_1_ represent the input, whereas C_3_ represent the output. The turns ratio of the transformer was set to (a) 1:72 and (b) 1:91. We can find the turns ratio in inductor (L_1_) as follows in Equation (2) [21,28].
(2)$$\sqrt{\frac{L_{1}\;Output}{L_{1}\;Input}}=N\;(N\;is\;turns\;ratio)$$

The output capacitance C_2_ of the electrode was approximately (a) 31.47 pF, (b) 71.955 pF; R_load_ represents the open-circuit resistance (100 MΩ) that drives the circuit [29]. The measured passive component values for the simulation processes were as follows: (a) 0.5BZT–0.5BCT (R_1_ = 3 MΩ, L_1_ = 333.35 mH, C_1_ = 826 pF, C_2_ = 31.47 pF, and C_3_ = 91.72 pF at 60 kHz, input voltage 5 V) and (b) 0.5BZT–0.5BCT doped with Sb_2_O_3_ 0.1 mol% (R_1_ = 3.4 MΩ, L_1_ = 333.35 mH, C_1_ = 881 pF, C2 = 71.955 pF, and C3 = 97.93 pF at 58 kHz, input voltage 5 V). In this matching circuit, the amplification factor for each frequency is called the step-up ratio. This step-up ratio gives a vibrational frequency of about 60 kHz for a pure 0.5BZT–0.5BCT material. At this 60 kHz, the step-up ratio showed a value of 73.01, and it was found that the simulation value also changed as the value of the resistor capacitor inductor changed. BZT–BCT with 0.1 mol% dopant showed a step-up ratio of 91.13 at 57 kHz with a lower vibrational frequency [24,30]. This is designed by simulation to affect the vibrational frequency and output characteristics according to the capacitor of the device. Also, the simulation gives you the predicted piezoelectric transformer step-up ratio.

As seen in Figure 6a,b, these are data measured after manufacturing the device. Based on each simulation data, as seen in Figure 6c, the frequency and output characteristics were measured around the vibrational frequency. It was confirmed that the values were similar to those of the simulation data. Pure 0.5BZT–BCT showed a step-up ratio of 73 at a vibrational frequency of about 60 kHz, and 0.5BZT–0.5BCT doped with Sb_2_O_3_ 0.1 mol% obtained a step-up ratio of 83 at a vibrational frequency of 58 kHz. The error between simulation and design measurements was approximately 0.1%.
(3)$$\mathrm{FoM}\;(\mathrm{Figure}\;\mathrm{of}\;\mathrm{Merit})=\frac{{\left(V_{step-up\;ratio}\right)}^{2}\;}{Quality\;factor\;Q}$$

The following Equation (3) is the formula for obtaining the figure of merit (FOM). Since the Q value depends on the amount of the capacitor, there are many variables. Therefore, FoM was measured by squaring the amplification factor. The quality factor Q value is required to calculate the FoM [31].
(4)$$Quality\;factor\;Q=\frac{\omega_{0}}{\omega_{c2}-\omega_{c1}}=\frac{\omega_{0}}{\beta }$$

The following Equation (4) is the quality factor Q, which shows the relationship between the center frequency and the cutoff frequency as follows ($\omega_{0}$: center frequency; $\omega_{c1}$, $\omega_{c2}$: cutoff frequency). What can be seen here is that the higher the FoM, the more suitable it is for device use.
(5)$$FoMpure=\frac{{\left(V_{step-up\;ratio}\right)}^{2}\;}{quality\;factor\;Q}=35.06$$
(6)$${FoM}_{{Sb}_{2}O_{3}}\;dopant=\frac{{\left(V_{step-up\;ratio}\right)}^{2}\;}{quality\;factor\;Q}=47.59$$

Therefore, as shown in Equations (5) and (6), the 0.5BZT–0.5BCT doped with Sb_2_O_3_ 0.1 mol% displays a FoM value that is 47.59 higher than the 0.5BZT–0.5BCT pure FoM value of 35.06, which confirms that soft doping is suitable for a piezoelectric transformer element.

Figure 7 shows the step-up ratio concerning vibrational frequency for two types of piezoelectric transformers: one made of 0.5BZT–0.5BCT pure material and the other doped with 0.1 mol% Sb_2_O_3_. In the case of the pure material, the step-up ratio is 73 at a vibrational frequency of 60 kHz, with a FoM of 35.06. However, when the 0.1 mol% Sb_2_O_3_ dopant is introduced into the 0.5BZT–0.5BCT material, the step-up ratio increases to 83.46 at a slightly lower vibrational frequency of 58 kHz, with a FoM of 47.59. In summary, the FoM is 35.06 for 0.5BZT−0.5BCT and 47.59 for 0.5BZT−0.5BCT doped with 0.1 mol% Sb_2_O_3_.

## 4. Conclusions

In this study, a Rosen-type lead-free piezoelectric transformer was manufactured by employing the ceramic material 0.5Ba(Zr_0.2_Ti_0.8_)O_3_–0.5(Ba_0.7_Ca_0.3_)TiO_3_. The 0.5BZT–0.5BCT piezoelectric transformer exhibits P_r_ = 0.5 µC/cm^2^ and E_c_ = 6.5 kV/cm. A piezoelectric transformer 0.5BZT–0.5BCT doped with Sb_2_O_3_ 0.1 mol% exhibits P_r_ = 1.2 µC/cm^2^ and E_c_ = 4.5 kV/cm. Meanwhile, the soft dopant has characteristics of increasing d_33_, k_p_, permittivity, as well as the P_r_ and the E_c_ of piezoelectric properties, respectively. This affects the output characteristics in the piezoelectric transformer. It is described the same as in the equation expressing the output characteristics of the piezoelectric transformer. However, since they are all made of interactions, increasing d_33_ does not have a mechanism for increasing Q_m_ [32]. Therefore, it was confirmed that the doping of 0.1 mol% of Sb_2_O_3_, which has the most appropriate properties, not only has good physical properties but also has a good effect on output characteristics. The characteristics in the piezoelectric transformer were studied at a vibrational frequency in the range 52–68 kHz. According to a comparison of the simulation and measured data, according to the simulation and comparison, the 0.5BZT−0.5BCT piezoelectric transformer has the maximum step-up amplification ratio (V_out_/V_in_ = 73) and vibrational frequency (60kHz). The 0.5BZT−0.5BCT piezoelectric transformer doped with Sb_2_O_3_ 0.1 mol% has the maximum step-up gain ratio (V_out_/V_in_ = 91.13) and vibrational frequency (58 kHz). This device has properties suitable for use as a piezoelectric transformer. Finally, through FoM, the potential for practical use was confirmed. The FoM of the 0.5BZT−0.5BCT piezoelectric transformer is 35.06 and, when compared to the FoM of 0.5BZT−0.5BCT doped with Sb_2_O_3_ 0.1 mol%, was 47.59. The piezoelectric transformer with 0.1 mol% soft dopant has a high FoM and a low vibrational frequency. Therefore, it was established that 0.5BZT–0.5BCT with a 0.1 mol% doping level was the suitable choice and it is more likely to be used in practice.

## Figures and Tables

**Figure 1 materials-16-06201-f001:**
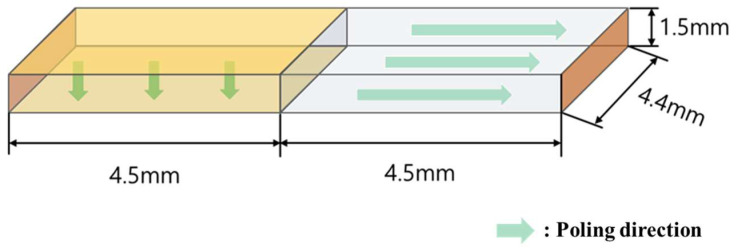
Rosen-type 0.5Ba(Zr_0.2_Ti_0.8_)O_3_–0.5(Ba_0.7_Ca_0.3_)TiO_3_ (0.5BZT–0.5BCT) piezoelectric transformer.

**Figure 2 materials-16-06201-f002:**
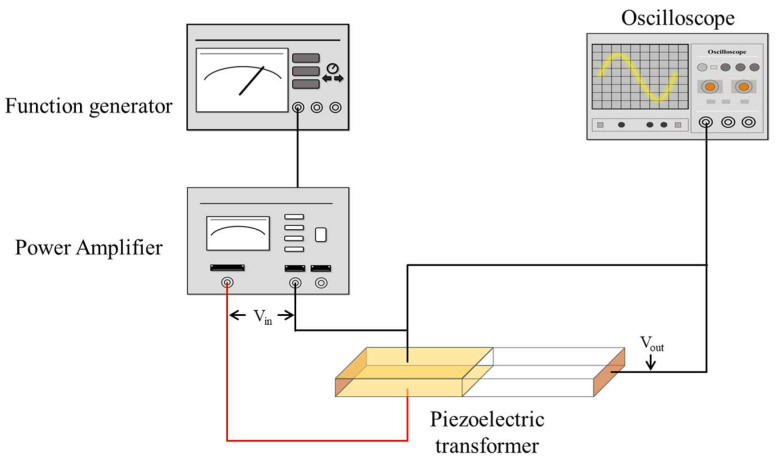
Schematic diagram of features set up for a piezoelectric transformer.

**Figure 3 materials-16-06201-f003:**
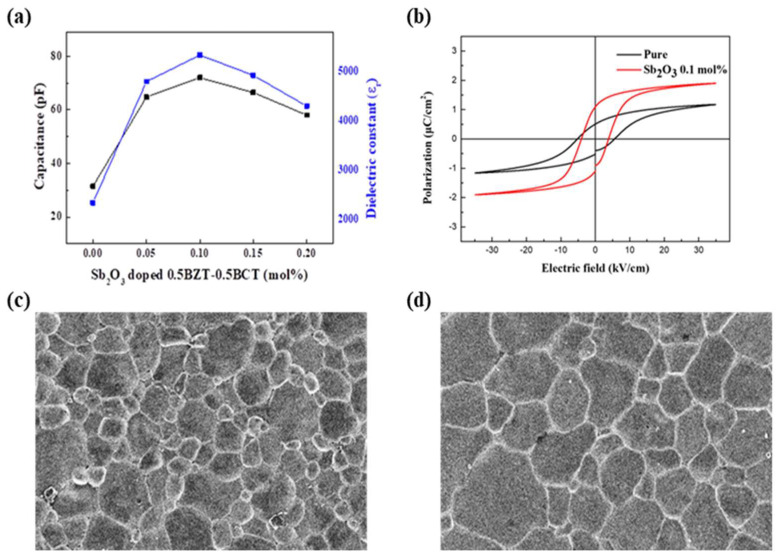
(**a**) Measured capacitance and dielectric constant according to the Sb_2_O_3_ dopant (0.0, 0.05, 0.1, 0.15, 0.2 mol%) 0.5BZT−0.5BCT at the normal temperature. (**b**) P-E loop of 0.5BZT−0.5BCT material and material doped with 0.1 mol% Sb_2_O_3_ at the normal temperature. (**c**) Image captured through scanning electron microscopy of ceramics composed of 0.5BZT−0.5BCT. (**d**) Scanning electron microscopy image 0.5BZT−0.5BCT doped with 0.1 mol% Sb_2_O_3_ ceramics.

**Figure 4 materials-16-06201-f004:**
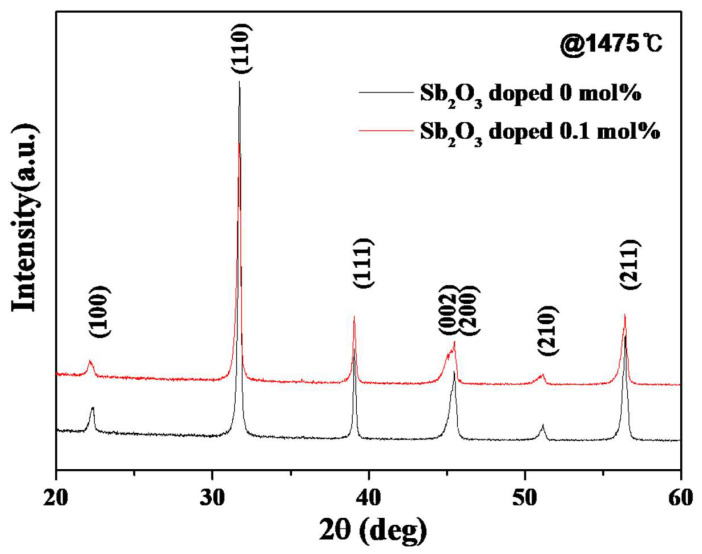
XRD patterns of 0.5BZT−0.5BCT ceramics, with and without doping with 0.1 mol% Sb_2_O_3_.

**Figure 5 materials-16-06201-f005:**
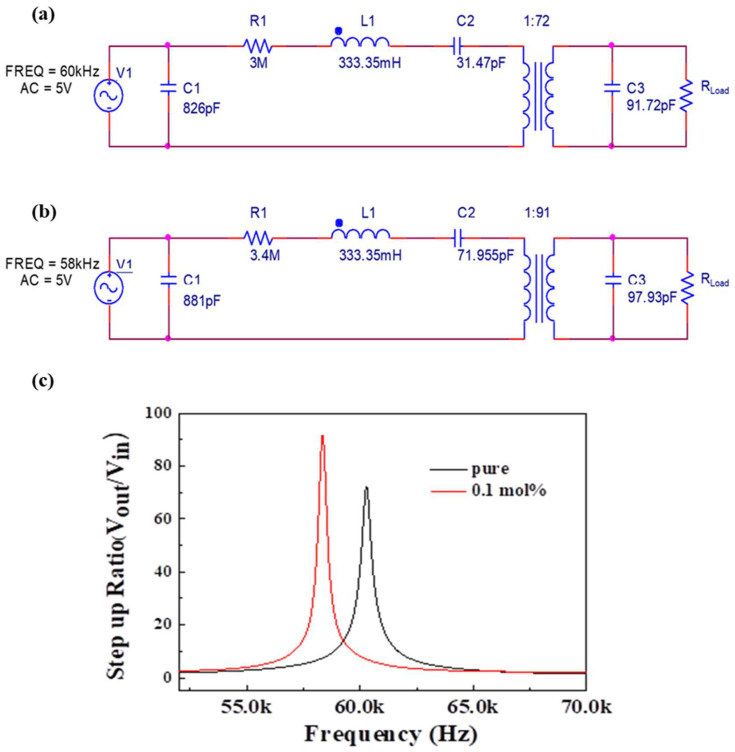
Matching circuit for a piezoelectric transformer for (**a**) 0.5BZT−0.5BCT, (**b**) dopant Sb_2_O_3_ 0.1 mol% 0.5BZT−0.5BCT, and (**c**) simulation of a step-up ratio of 0.5BZT−0.5BCT pure and Sb_2_O_3_ 0.1 mol% dopant.

**Figure 6 materials-16-06201-f006:**
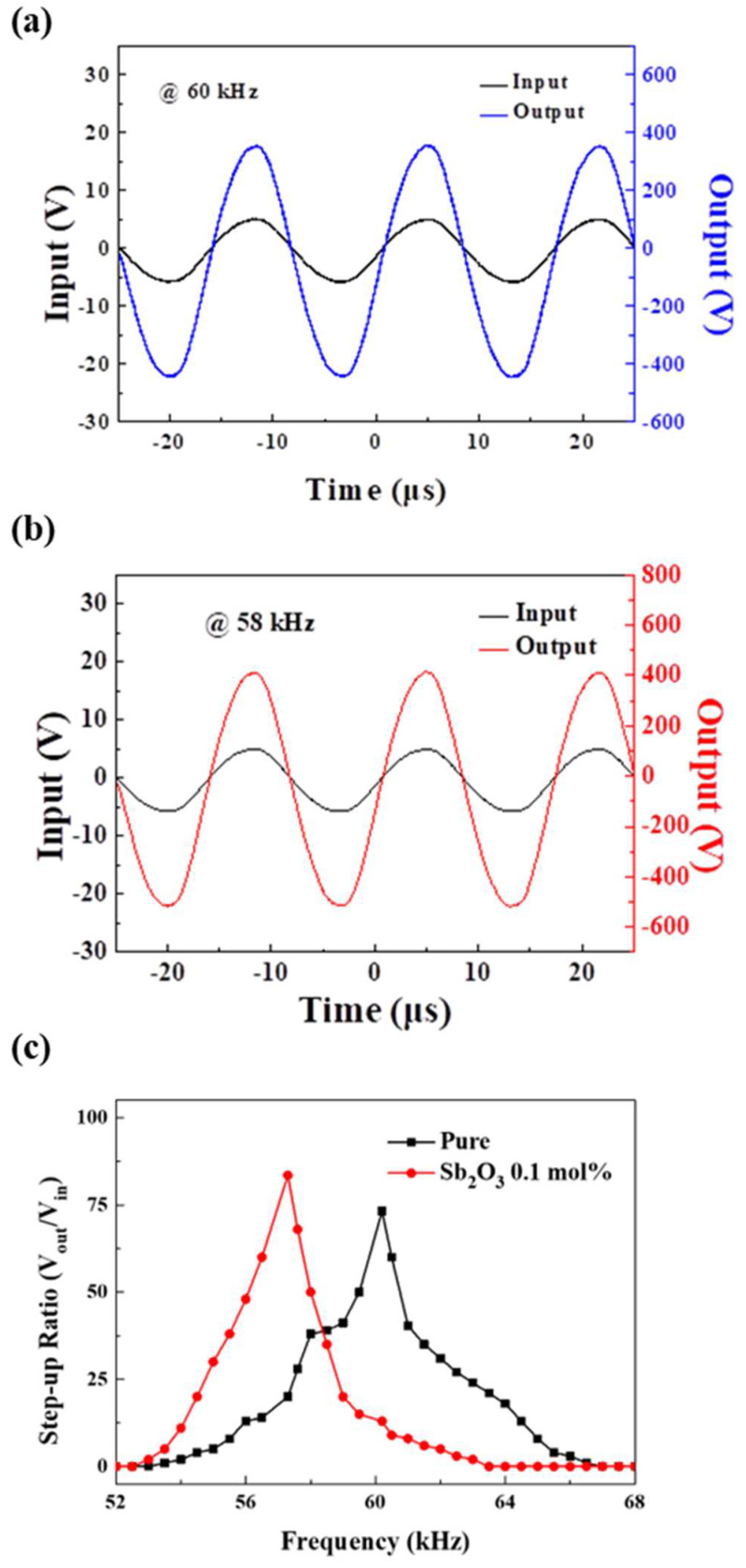
(**a**) Output voltage at the vibrational frequency of 60 kHz of the piezoelectric transformer according to the input voltage in a pure material at normal temperature, (**b**) output voltage at the vibrational frequency of 58 kHz of the piezoelectric transformer according to the input voltage doped with 0.1 mol% Sb_2_O_3_, and (**c**) measurement of the step-up ratio according to the frequency of the 0.5BZT−0.5BCT pure and Sb_2_O_3_ 0.1 mol% dopant (52 kHz~68 kHz) at the normal temperature.

**Figure 7 materials-16-06201-f007:**
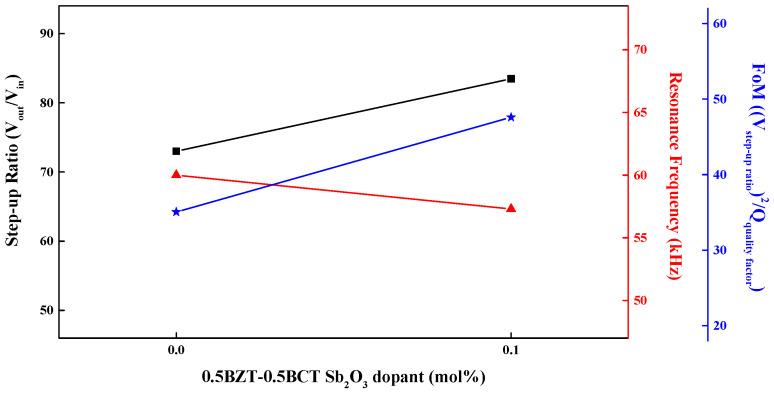
Step-up ratio according to the vibrational frequency in the piezoelectric transformer with 0.5BZT–0.5BCT pure and 0.1 mol% dopant Sb_2_O_3_.

**Table 1 materials-16-06201-t001:** Piezoelectric properties of a 0.5BZT−0.5BCT pure material and a material doped with 0.1 mol% Sb_2_O_3_.

	Properties	d_33_	ε_r_	Q_m_	k_p_	k_33_	k_31_	Step-Up Ratio
Material		(pC/N)	(1 kHz)	(1 kHz)	(%)	(%)	(%)	(V_out_/V_in_)
Pure		470	2860	68	42.4	51.1	68.1	73.01
Sb_2_O_3_ 0.1 mol%		581	4040	87	55.3	56.2	62.1	91.13

## Data Availability

The data presented in this study are available on request from the corresponding author.

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
