# Peer review of "Rosen-Type Piezoelectric Transformers Based on 0.5Ba(Zr0.2Ti0.8)O3–0.5(Ba0.7Ca0.3)TiO3 Ceramic and Doped with Sb2O3"

_materials, 2023, doi:10.3390/ma16186201_

Round 1

Reviewer 1 Report

The manuscript demonstrated the Sb2O3-doped alloyed ceramic for piezoelectric transformers. But the results are not convincing so that I would recommend against its publication unless the major revision was made. Details have been listed below.

1. Structural information is absent in the manuscript. The morphological and elemental analysis should be carried out by scanning electron microscopy and energy dispersion spectroscopy at least.

2. Doping levels should be explored and discussed in detail. What is the standard that the authors chose the 0.1% doping level as the example, and why?

3. The authors just compared the properties of the materials without doping and with 0.1% doping. Moreover, Figure 5 just show 2 points in each data line, which is quite confusing and misleading. Please show at least 4~5 different doping levels to release the trends.

Author Response

Author to respond reviewer_1, and revised manuscript

Reviewer 2 Report

This paper studies the sensitivity of Rosen type piezoelectric transformers, based on lead-free piezoelectric ceramic, to Sb2O3 doping. The introduction is correctly documented. Details about the fabrication of piezoelectric discs are sufficient. The setups used to measure the different properties of the materials are partially described. A simple equivalent circuit is used to simulate the behavior of the system. The interest of this simulation is unclear and seems to bring no real interest to this study. The sensitivity of the material and system when varying the doping is well established but seems to represent a modest scientific concern. In addition, further points could be performed :

- Terms in eq 1 a have to be defined.

- Ref 15, used to define eq 1, is not correctly numbered.

- Figure 2 (second figure 1) is confusing as Vin appears twice. The reference of Vin and Vout should represented.

- Accuracy of the material properties measurement ( dielectric and electromechanical) could be added.

- There is confusion in the numbering of the figure (there are two figures 1).

- The measurement setup and experimental conditions to characterize P(E) loops and Capacitance must be introduced and detailed.

- The choice of the equivalent circuit to model the piezoelectric transformer must be justified.

- The method used to determine the turns ratio of the transformer should be detailed.

- The relevance of this simulation should be discussed.

Author Response

Author to respond reviewer_2, and revised manuscript

Reviewer 3 Report

Comments in attachment

Author Response

Author to respond reviewer_3, and revised manuscript

Reviewer 4 Report

The manuscript entitled “Rosen-type piezoelectric transformers based on 0.5Ba(Zr0.2Ti0.8)O3–0.5(Ba0.7Ca0.3)TiO3 ceramic and doped with Sb2O3” presents values of the piezoelectric characteristics for the pure and, unfortunately, one (0.1 mol%) material. I think a potential reader would be more satisfied if they could see the quoted values for the other values of doped materials, i.e., 0.05, 0.15, and 0.2 mol% of Sb2O3. Despite the lack of the values of these quantities, which I hope will be supplemented in the case of publication of the manuscript as a scientific article, the presented manuscript is well prepared, clearly described, and very interesting. Thus, I think that it should be published in the Journal after the amendments.

1.      Page 2, lines 63 and 65 (Eq(1)): All shortcuts/abbreviations should be explained before the first used, as “Q_m” (mechanical quality factor), “L_out” (output part length), “t” (bar thickness), “s_D_33 / 11” or “s_E_33 / 11” (components of elastic compliance at constant electric displacement or electric field, respectively).

2.      Page 2, line 89: (i) Add information about the temperature of the milling process, and characteristics of the balls used (size, materials, etc.). (ii) Add information about the temperature of the drying process.

3.      Page 3, lines 96-98 and 108-109: There is no information about temperatures of the measurements, regimes used (cooling or heating, etc.), etc.

4.      All figures, except Fig. 1 (page 3): Renumber the figures presented: Fig.1 (page 4) – it should be Fig. 2; Fig. 2 (p. 4) – it should be Fig. 3, etc.

5.      Figures 3(a) and 3(b) – Add errors of the measured quantities or give the errors as “%” in the text.

6.      Figures 3(a)-(b), 5(a)-(c), 6, and Table 1: Add information about the temperature of the measurements.

7.      Table 1: Did the authors measure the characteristics shown in Table 1 (p. 4) for all materials (i.e., with various Sb2O3 dopant values)? I think that these values should be also present.

8.     Page 5, lines 172-173, Figs. 4(a)-(b), 5(c): The errors for the measured values/calculated values should be given.

9.      Page 6, line 179: The authors have given the frequency of 57 kHz for the step-up ratio of 91.13. I think that the frequency value should be higher! Compare Fig. 5(c) and 4(c) – the red curves, and page 8, line 242 (about 58 kHz)!

10.  Page 7, lines 188-192, and page 7, lines 217-218: Parts of the text are repetitions of the Figure 5 caption (page 6, lines 183-187) and Figure 6 caption (page 7, lines 215-216), respectively. These parts have to be rewritten!

11.  Figure 6 (p. 7): Figure 6 gives no additional data/information (the data/values are given within the text). If the authors present in Fig. 6 values of these quantities for all materials, i.e., for 0.05, 0.1, 0.15, and 0.2 mol%, then it will be more suitable for the conclusion!

12.  Page 8, line 220: I think that the resonant frequency for the Sb2O3 0.1 mol% doped material is about 58 kHz (see red curves in Figs. 4(c) and 5(c), and text – page 6, line 179, and page 7, line 198!

13.  The minor errors and misprints found: (i) page 1, lines 43-44: the sentence “The acceptor doping crates hard while donor doping creates soft.” As a little laconic. It should be rewritten; (ii) page 3: the authors used various sizes of the font (compare parts of the text below and above Fig 1); (iii) page 5, line 175 and page 7, line 197: there should be “(…) a resonant frequency of about (…)”; (iv) Data shown in Figs. 4(c) and 5(c) can be shown in one common figure; (v) Equations (4) and (5): the values of the “V_step-up ratios” and “Quality factors Q” (the numerical values given between the equality signs) are not necessary. Errors should be added; (vi) page 8, line 226: the double commas used.

Some parts of the text are in laconic style. There are some repetitions. 

Author Response

Author to respond reviewer_4, and revised manuscript

Round 2

Reviewer 1 Report

I recommend the publication of the revised version.

Reviewer 2 Report

The modifications proposed are in good agreement with the comment previously made. However, the details about the measurement of the P(E) loops are still missing. Typically, the method used to measure the polarization is missing. Details about the accuracy of each material's characteristics could bring an interesting worth to this work.

Could be performed

Reviewer 3 Report

The authors have responded to the comments/ questions, but it is insufficient to answer previous questions/comments. Please correct and elaborate on the answer according to the following comments.

1.      "The introduction fails to look at what has been done in the past, the missing link, and what is being proposed to fill the missing kink. The introduction does not include the essence of the work. A statement of the research problem and the justification for the study should also be included in the introduction."

It means, "What are the novelties of the work as compared to prior studies? Or what made the authors conduct this study?

2.      The experiment method was not described sufficiently, so the approach used cannot be understood. The sub-section 2.2 was just a repetition from section 2.1, so this should differ clearly between discs and transformers. The characterizations should be written in detail.

3.      "The discussion part should be improved."

a.      "Since the authors pointed to the fabrication of Sb2O3-soft doped 0.5BZT-0.5BCT materials, proper results showing that the doping has succeeded should be included."

The authors only give SEM images without any explanation/discussion. Besides, SEM images do not sufficiently prove that the doping has been successfully conducted (except some studies reported this issue).

The authors should include data, like XRD, for example, or similar things, to show their success in doping to demonstrate that their fabrication was successful.

b.      "The roles of Sb2O3-soft doped in the 0.5BZT-0.5BCT have not been included in the discussion. Why the 0.1% Sb2O3 revealed the optimum/ best piezoelectric transformer characteristics should be explained."

The authors have not discussed or provided answers regarding the role of doping in improving the piezoelectric transformer characteristics. It should be related to the doping roles. For instance, if the doping changes the crystal structure, what is the correlation between the crystal structure and electrical properties? Or, it can be related to other properties that may impact the electrical characteristics.

c.      "Please check the Figures' order because the Figures mentioned did not match the numbers."

The order in Figures has not been checked and fixed. There are 2 Figures 1, so it changes their order.

d.      In lines 245 – 251, pardon why the sentences must be made point by point. It feels weird. Please make a paragraph.

1.      The conclusion part should be elaborated based on its research purposes.

2.      The similarity check by Turnitin showed that the paper's similarity is still high at about 30%. Please revise it so that it can decrease.
